# How Accurately Can Wearable Sensors Assess Low Back Disorder Risks during Material Handling? Exploring the Fundamental Capabilities and Limitations of Different Sensor Signals

**DOI:** 10.3390/s23042064

**Published:** 2023-02-12

**Authors:** Cameron A. Nurse, Laura Jade Elstub, Peter Volgyesi, Karl E. Zelik

**Affiliations:** 1Department of Mechanical Engineering, Vanderbilt University, Nashville, TN 37212, USA; 2Department of Biomedical Engineering, Vanderbilt University, Nashville, TN 37212, USA; 3Institute for Software Integrated Systems, Vanderbilt University, Nashville, TN 37212, USA; 4Department of Physical Medicine & Rehabilitation, Vanderbilt University, Nashville, TN 37212, USA

**Keywords:** risk assessment, lifting biomechanics, ergonomics overexertion injuries, work-related musculoskeletal disorders

## Abstract

Low back disorders (LBDs) are a leading occupational health issue. Wearable sensors, such as inertial measurement units (IMUs) and/or pressure insoles, could automate and enhance the ergonomic assessment of LBD risks during material handling. However, much remains unknown about which sensor signals to use and how accurately sensors can estimate injury risk. The objective of this study was to address two open questions: (1) How accurately can we estimate LBD risk when combining trunk motion and under-the-foot force data (simulating a trunk IMU and pressure insoles used together)? (2) How much greater is this risk assessment accuracy than using only trunk motion (simulating a trunk IMU alone)? We developed a data-driven simulation using randomized lifting tasks, machine learning algorithms, and a validated ergonomic assessment tool. We found that trunk motion-based estimates of LBD risk were not strongly correlated (r range: 0.20–0.56) with ground truth LBD risk, but adding under-the-foot force data yielded strongly correlated LBD risk estimates (r range: 0.93–0.98). These results raise questions about the adequacy of a single IMU for LBD risk assessment during material handling but suggest that combining an IMU on the trunk and pressure insoles with trained algorithms may be able to accurately assess risks.

## 1. Introduction

Low back disorders (LBDs) are a leading occupational health issue, and commonly due to overexertion. LBDs account for almost 40% of work-related musculoskeletal disorders in the U.S. [1], resulting in financial and personal burdens due to healthcare costs, time off work, lost productivity, physical pain, and psychological distress. Individuals working in manual material handling jobs are at an elevated risk of LBDs due to overexertion experienced by the back during repetitive lifting and bending [2,3]. Overexertion injuries result from the accumulation of microdamage caused by repeated loading of musculoskeletal tissues (e.g., muscles, tendons, ligaments, discs), and are consistent with a mechanical fatigue failure process [4,5,6].

Ergonomic interventions have reduced work-related musculoskeletal disorder prevalence [7,8,9] by reducing physical strain on workers [10,11]. The hierarchy of controls is a useful framework for implementing interventions after risks are identified [9].

Ergonomic assessments identify injury risk and play a critical role in informing and prioritizing where ergonomic interventions are needed. Several ergonomic assessment tools have been developed to identify LBD risk resulting from manual lifting tasks. The Lifting Fatigue Failure Tool (LiFFT) is noteworthy because it is based on fatigue failure principles [12] and estimates cumulative damage of the low back. LiFFT works by counting the number of lifting repetitions and using peak load moment (an indicator of back strain due to the object lifted) to estimate damage inflicted on the person’s back each lift [12]. Gallagher et al. used mechanical fatigue failure principles to develop a relationship between lumbar moment and tissue damage per cycle, which underlies LiFFT [12]. LiFFT was validated against existing epidemiological databases [13,14] containing the peak load moment for each lift, the number of lifts performed per day, and the incidence of LBD for workers in diverse occupations [12,13].

However, ergonomic assessments can be time-consuming, costly, and inconsistent in practice. For instance, approximately one in three assessments has been found to contain errors, and in 13% of the assessments, the errors were substantial enough to invalidate the evaluation [15]. To perform an ergonomic assessment with LiFFT, a trained safety professional observes an individual worker during a shift, manually recording the weight of each object lifted, the horizontal distance from the object to the lumbar spine (or hip), and the number of repetitions of this task [12]. Due to the time-intensive nature of these observations, continuous and personalized monitoring is often impractical or cost prohibitive. This means that rather than assessments occurring over a long duration and involving a variety of tasks across multiple workers, conclusions are drawn from limited observations and may not be generalizable across a wide range of workers or tasks.

There is an opportunity to develop more efficient and consistent ways to assess ergonomic injury risk over a range of workers and workplace environments by providing automated monitoring of overexertion risk (e.g., to the low back). Wearable sensors are one type of emerging technology that could improve ergonomic assessments by monitoring individuals remotely and for long durations, allowing for personalized injury risk assessment to inform ergonomic interventions. This could reduce the time and effort burden on safety professionals, while simultaneously enabling more widespread risk assessment.

Wearable sensor systems using inertial measurement units (IMUs) are already being explored and adopted in industry to complement or expedite ergonomic assessment [16,17]. IMU systems provide kinematic information [18,19,20] and are beneficial for ergonomic assessments that seek to capture data related to high-risk postures and bending frequency. There are some reports from industry indicating that IMU systems are useful in identifying and reducing musculoskeletal injury risks [16,17], and quantitative claims are now being made by device manufacturers about the efficacy and capabilities of their IMU systems for injury reduction. However, there is a dearth of formal, independent, or peer-reviewed studies that have evaluated these IMU systems, or the scientific basis underlying their risk assessment algorithms.

One known limitation of IMU systems is that they do not measure the weight of an object lifted, which may limit their ability to identify injury risk due to physical overloading or overexertion [21]. While IMU systems can identify when an individual has performed a forward bend, they are unable to determine the weight of the object lifted or if an object has been lifted at all. This limitation is significant because musculoskeletal loading is dependent on forces and moments (kinetics), which require knowledge of the weight of an object lifted. Manufacturers of these IMU systems have tried to implement workarounds, such as manually inputting object weight, assuming a nominal object weight, or developing algorithms that attempt to estimate object weight indirectly based on movement patterns or other heuristics. However, the validity and efficacy of these workarounds is unknown. Fundamentally, it remains unclear to what degree we should expect IMU systems to be capable of assessing overexertion injury risk to the low back.

Combining pressure-sensing insoles with IMU sensors has the potential to overcome this limitation and to provide more accurate ergonomic assessment of lifting tasks. We previously found that lumbar moment, an indicator of low back loading, can be estimated 33% more accurately with combined signals from a trunk IMU and pressure insoles than with trunk IMU signals only, based on an analysis using idealized wearable sensor signals [21]. However, material (e.g., tissue) damage is exponentially related to the peak load applied [4,5,6], meaning that modest errors in lumbar moment estimates [21,22,23] can result in much larger errors in damage estimates, affecting estimates of injury risk. These prior findings [21] suggest that improved accuracy from combining pressure-sensing insoles and an IMU might meaningfully improve LBD risk assessment in the workplace. However, this proposition requires further research and validation, and important open questions remain, including:How accurately can we estimate LBD risk with a trunk IMU combined with pressure insoles?How much greater is this risk assessment accuracy than using a trunk IMU alone?

The objective of this study was to address the two open questions above. To maximize the generalizability of our results and study contributions, we used a data-driven simulation approach to characterize and explore how accurately we could classify LBD risk using only trunk motion (e.g., from a single trunk IMU) versus using trunk motion and forces under the feet (e.g., from a single IMU combined with pressure insoles).

The first question is important because prior literature [21,22,23] only quantified the accuracy of a biomechanical indicator of loading on the lower back (i.e., lumbar moment). Ultimately, what safety professionals care about is how accurately they can classify injury risk (not just estimate time-series biomechanical variables such as lumbar moment). LBD risk assessment accuracy has not yet been studied for trunk IMU and pressure insole systems. Furthermore, it is currently unclear what level of biomechanical load accuracy is good enough for ergonomic assessment in practice, particularly given the nonlinear relationships between musculoskeletal load, damage, and injury risk [6,12,24].

The second question is important because it is currently unclear whether improved accuracy—which we expect using additional information from the pressure insoles—is substantial enough to justify the extra cost and complexity of adding pressure sensors into a user’s shoes. It is possible that processing IMU data alone may be sufficient for obtaining reasonably accurate estimates of LBD risk over a full workday (e.g., hundreds of lifts). This may be true even if the IMU-estimated accuracy is worse than the combined trunk IMU and pressure insole accuracy on a lift-by-lift basis.

There were three possible conclusions from our study:

**Possibility** **1:**Capturing under-the-foot force data (e.g., from pressure insoles) is unnecessary because trunk motion alone (e.g., from an IMU) can provide reasonably good accuracy in estimating LBD risk over the course of a workday.

**Possibility** **2:**Trunk motion data alone are insufficient to accurately estimate LBD risk, but adding under-the-foot force data results in a substantial improvement and reasonably good accuracy in LBD risk estimation.

**Possibility** **3:**Neither trunk motion nor trunk motion combined with under-the-foot fore data seems adequate to accurately assess LBD risk over a workday, suggesting that alternative risk assessment tools may be needed.

Any one of these conclusions would be interesting, insightful, and actionable with respect to the potential application of wearable sensors to automate ergonomic risk assessment for the low back.

## 2. Materials and Methods

The study was accomplished in five stages. First, we analyzed an existing dataset [21], which contains full-body 3D (three-dimensional) motion capture and bilateral ground reaction force data from 10 individuals performing a range of lifting tasks to obtain lab-based estimates of lumbar moment (an indicator of low back loading). Second, we trained a machine learning algorithm to estimate peak lumbar moment during each lift using only trunk motion signals (which we refer to as the *Trunk* signals and algorithm) and then using trunk motion and force under the feet (*Trunk+Force* signals and algorithm). For this analysis, we used idealized wearable sensor signals, meaning that data from laboratory instrumentation were converted into signals reasonably obtained from wearable sensors (rationale detailed below). Third, we converted the peak *lumbar* moments into peak *load* moments to use as inputs to the ergonomic assessment tool, LiFFT. Fourth, we simulated 10,000 material handling workdays based on randomly selected lifts and used LiFFT to compute LBD risk. Fifth, we evaluated the accuracy of the Trunk and Trunk+Force approaches by comparing their estimates of LBD risk during each simulated workday to lab-based (ground truth) estimates of LBD risk.

### 2.1. Data Collection and Processing

We reanalyzed a subset of an existing dataset [21] in which 10 participants performed manual material handling tasks. For details of the experimental protocol and reasoning, see Matijevich et al. [21]. We selected the 50 tasks involving symmetrical lifting of objects of different weights (5–23 kg) from different heights (40–160 cm above the ground) to simulate jobs of varying physical demands. Ground reaction forces and the center of pressure under each foot were collected with force plates (Advanced Mechanical Technology Inc., Watertown, MA, USA). Full-body kinematics were collected simultaneously from a 3D motion capture system (Vicon, Oxford, UK). Body segmental and joint kinematics were estimated from the motion capture data and rigid-body inverse kinematics in Visual3D (C-Motion, Germantown, MD, USA) software. Lab-based lumbar moment was estimated using bottom–up inverse dynamics, which prior studies have found to be in good agreement with top–down inverse dynamics during lifting [22,25]. We refer to these lumbar moments as lab-based (or ground truth) estimates since we used comprehensive 3D kinetic and kinematic data from laboratory instrumentation. In contrast, the Trunk and Trunk+Force lumbar moment estimates (detailed below) only used a small subset of signals to reflect wearable sensor measurement capabilities.

All participants gave written informed consent to the original protocol, which was approved by the Institutional Review Board at Vanderbilt University (IRB #141697).

### 2.2. Algorithm Development

We trained a gradient boosted decision trees machine learning algorithm to estimate lumbar extension moment using methods similar to Matijevich et al [21]. This algorithm estimates the target metric by building a series of decision trees in stages. Each stage seeks to optimize the final prediction by estimating the residual error of the predictions from the previous stage. The final algorithms produced the lumbar moment estimates based on a series of approximately 100 trees. We used scikit-learn library and Amazon SageMaker, a cloud-based machine learning platform, for algorithm development, model training, and evaluation.

The lumbar extension moment during lifting was chosen as the target metric because it is an indicator of loading on the lower back and can be adapted to be an input to an established ergonomics assessment tool, LiFFT (see Section 2.3), to estimate cumulative damage and injury risk to the low back [6,12]. We estimated the time-series target lumbar moment from lab-based data [21], as detailed in Section 2.1. This resulted in a lookup table for each participant containing a lab-based peak lumbar moment estimate for each lifting task completed.

We used idealized wearable signals as inputs to train each algorithm. Idealized wearable signals refer to motion lab data converted into time series signals that can be feasibly obtained from wearable sensors [21,26]. Using idealized wearable signals enabled us to address the questions outlined in the Introduction in the most fundamental and generalizable way, without being limited by the quality of existing wearable sensor hardware or their calibration methods, and without narrowly evaluating a single product. When using actual wearable sensors—specifically pressure insoles—task- and hardware-specific calibration methods are typically required to obtain accurate and reliable force estimates [27]. However, developing these calibrations is complex and time-consuming. Therefore, our approach was to first evaluate and understand the expected necessity of pressure insole data in this current study using idealized wearable signals. The results of this study would inform which types of wearable sensors are needed and provide the basis for deciding whether to invest time into developing custom pressure insole calibrations.

We trained two separate algorithms using two different sets of signals: the first algorithm used Trunk signals as inputs, and the second algorithm used Trunk+Force signals as inputs. IMU sensors estimate orientation in the global coordinate system. IMU angle estimates are strongly correlated with angles obtained from 3D motion capture [28,29]. Therefore, Trunk input signals were composed of 3 directional orientation angles, accelerations, and angular velocities in the global coordinate system, which were approximated using 3D motion capture. Pressure insoles estimate ground reaction force and the center of pressure in the foot’s coordinate system. These have been found to be correlated with force plate data [30,31]. Thus, Force input signals were composed of a 1D force vector and the center of pressure in the mediolateral and anteroposterior directions, approximating signals that can be obtained from pressure insoles. The center of pressure from the force plate was transformed into the foot’s coordinate system, and the 1D force vector under each foot was obtained by projecting the 3D ground reaction force from the force plate onto a vector normal to the foot.

To train each algorithm, we used cross validation by participant [32]. In other words, the data from 9 participants were used to train the algorithm, and then the trained algorithm was applied to the 10th participant’s data to estimate lumbar moment [32]. This process was repeated for all 10 participants, resulting in two lookup tables per participant, one for each wearable signal condition (Trunk, Trunk+Force), containing the lumbar moment estimates for each lifting task. Cumulative damage and injury risk estimated from fatigue failure models are driven by high load magnitudes [4,5,12]. Therefore, to improve estimates of the larger lumbar moments and avoid overfitting lower lumbar moments (e.g., during upright standing), we applied a minimum threshold of 100 Nm on the target metric when training each algorithm. As such, only data points where lumbar moments were above 100 Nm were included in the training dataset, prioritizing trained algorithm performance at the high lumbar moment magnitudes that occur during lifting.

Peak lumbar moment lookup tables were then converted into peak load moment lookup tables. Peak load moment is another low back load metric and the specific input required by LiFFT [12,24]. This conversion from lumbar moment to load moment was achieved by subtracting out moment contributions from the upper body orientation and acceleration, and calculations are explained in detail below.

### 2.3. Converting Lumbar Moment into Load Moment

The peak load moment ($M_{Load,i}$) was calculated during each *i*th lift [12,24] to assess LBD risk using LiFFT. Load moment ($M_{Load}$) is defined as the time-series moment created about the lumbar spine by the object lifted and calculated by multiplying the weight of the object by the horizontal distance from the object to the lumbar spine (L5/S1). Peak load moment refers to the maximum load moment during a lift. However, the machine learning algorithm from Matijevich et al [21]. and summarized in Section 2.2 was trained to estimate *lumbar moment* ($M_{Lumbar}$). Lumbar moment includes moment contributions due to the weight of the lifted object ($M_{Load}$), the linear acceleration of the upper body and object ($M_{Linear}\;$), and the rotational acceleration of the upper body ($M_{Rotation}\;$). Thus, to estimate load moment, we performed the following calculation:(1)$$M_{Load}=\;M_{Lumbar}-M_{Linear}-M_{Rotation}$$
(2)$$M_{Linear}=m_{HAT}\left(a+g\right)L+m_{box}aL$$
(3)$$M_{Rotation}\;=\;I\alpha$$

*L* is the horizontal distance from the lumbar spine to the center of the mass of the head–arms–trunk (HAT), estimated using anthropometric tables [33]. The dataset did not include markers on the box; thus, we assumed that the object lifted was accelerating at the same rate as the upper body and was being held close to the center of the mass of the upper body. $m_{HAT}$ is the mass of the HAT, estimated using anthropometric tables [32]. $m_{box}$ is the mass of the object being lifted in the analyzed trial. a is the acceleration of the upper body, estimated with the idealized trunk IMU signals. I is the moment of inertia of the HAT, estimated using anthropometric tables [32], assuming that the HAT is a rectangular prism. α is the angular acceleration of the upper body, estimated using the idealized trunk IMU signals. These assumptions seemed reasonable for the lifting tasks performed because the mass of the upper body is much larger than the mass of the box, meaning that the contribution of the box to $M_{Linear}$ is expected to be small by comparison. See Section 4.4 for more discussion about this topic, including post hoc analyses that provided additional support for these assumptions being reasonable.

### 2.4. Simulated Workdays

We explored 1000 simulated workdays per participant using LiFFT and empirical data from the peak load moment lookup tables. A simulated workday consisted of 800 to 2000 randomly selected lifts from the lookup table. For each lift (i), we extracted the peak load moment ($M_{Load,\;i}$) from the lookup table (lab-based, Trunk, or Trunk+Force), then used it as an input to LiFFT [12,24]. The cumulative damage over all lifts (D) was computed and then used to estimate the LBD risk for a simulated workday [12,24].
(4)$$D=\frac{1}{902416}\sum_{i=1}^{n}e^{0.038\left(M_{Load,\;i}\right)+\;0.32}$$

LBD risk (R, Equation (5)) was computed using a binary logistic regression equation previously developed using epidemiological databases with injury prevalence categories [13,14]. The term LBD risk refers to the probability of being in a high-risk job (not the probability of someone being injured). This definition originates from the epidemiological databases used to validate LiFFT [13,14].
(5)$$R=\frac{e^{Y}}{1+e^{Y}}$$
(6)$$Y=1.72+log_{10}\left(D\right)$$

For each simulated workday, LBD risk was computed separately from the lab-based, Trunk, and Trunk+Force lookup tables. In total, this simulated workday analysis resulted in a 1000 × 3 table of LBD risk results for each participant, where each row corresponded to one simulated workday, and the three columns were three conditions (lab-based, Trunk, and Trunk+Force). An overview of the lab-based data analysis, algorithm development, and LBD risk assessment is provided in Figure 1.

### 2.5. Evaluation

To evaluate the accuracy of LBD risk estimates from Trunk and Trunk+Force algorithms relative to lab-based LBD risk estimates (Figure 1), we calculated the root mean square error (RMSE) for each participant across all workdays. We then calculated the between-participant average RMSE.

To help answer the question of whether Trunk+Force signals improve injury risk assessment accuracy relative to Trunk signals alone, we performed a statistical analysis to compare RMSE results between conditions. First, a Kolmogorov–Smirnov test was used to confirm normal distribution of the RMSE results. Subsequently, a dependent (paired-samples) *t*-test was performed to compare RMSE from Trunk vs. Trunk+Force algorithms. The *p*-value and effect size (Cohen’s *d*) were calculated. The Pearson correlation coefficients (r) between the lab-based and Trunk LBD risks and between the lab-based and Trunk+Force LBD risks were also computed.

To gain insight into the practical significance of different levels of accuracy, we computed an additional summary metric: the percentage of simulated workdays where Trunk or Trunk+Force LBD risk estimates were within ±10% of the full-scale lab-based LBD risk assessment. For instance, if lab-based (ground truth) LBD risk was 65%, then Trunk or Trunk+Force estimates between 55% and 75% would be within ±10% based on this metric.

## 3. Results

Examples of idealized wearable Trunk and Force data during lifting are shown in Figure 2. Using Trunk+Force signals, LBD risk was estimated with an average RMSE of 4.7% compared with lab-based estimates (Figure 3, Table 1). LBD risk estimated with Trunk+Force signals was within ±10% of the lab-based estimates for 87.2% of all simulated workdays (Figure 4, Table 2).

Using Trunk signals only, LBD risk was estimated with an average RMSE of 15.8% (Figure 3, Table 1), which was a significantly larger error compared with using Trunk+Force signals (*p* < 0.01, *d* = 4.2). LBD risk estimated with Trunk signals was within ±10% of the lab-based estimates for 50.9% of the simulated workdays (Figure 4, Table 2).

The Trunk+Force estimates of LBD risk were strongly correlated with lab-based estimates, with participant-specific correlation coefficients ranging from *r =* 0.93 to 0.98 (Figure 5B, Table 3). The Trunk LBD risk estimates were weakly or moderately correlated with the lab-based estimates, with participant-specific correlation coefficients ranging from 0.20 to 0.56 (Figure 5A, Table 3).

## 4. Discussion

### 4.1. Summary

We found that combining trunk motion with under-the-foot force data meaningfully improved the accuracy of LBD risk estimates compared with using only trunk motion data. Trunk estimates of LBD risk were not strongly correlated with ground truth LBD risk estimates obtained using lab-based instrumentation (Figure 5A). Trunk motion may inform injury risk due to high-risk postures but did not accurately estimate or correlate with LBD risks due to repetitive lifting and musculoskeletal loading. For an individual workday, it was a coin flip (50% chance) as to whether the Trunk-based estimate would be within 10% of the lab-based LBD risk; however, by adding under-the-foot force data, this increased to nearly 90% (Table 2, Figure 4). Furthermore, Trunk+Force estimates correlated strongly with LBD risk across a large range of workday intensities (Figure 5B). Force under each foot provides critical information that can distinguish if a person is lifting a heavier vs. lighter object—which in turn improves LBD risk estimates. In practice, one way this force information can be obtained is with pressure-sensing insoles. These results highlight the potential for combining pressure-sensing insoles with a single IMU on the trunk to accurately perform remote and/or automated ergonomic assessment of LBD risks during lifting-intensive material handling jobs.

### 4.2. Insights on LBD Risk Accuracy and Wearable System Complexity

A key contribution of this study is that the analysis and simulations provided insight into which sensor signals are needed to accurately assess LBD risks. As discussed in the *Introduction*, it was unknown what level of biomechanical (e.g., lumbar moment) accuracy would be sufficient for workplace risk assessment. Furthermore, it was difficult to determine what might be good enough until we understood the impact on LBD risk estimates, which we quantified for the first time in this study. Our interpretation of the results is that Trunk motion signals alone are not sufficient to provide accurate estimates or reliable indicators of LBD risk when assessed using LiFFT. In contrast, the Trunk+Force estimates are promising because they provide more accurate estimates of LBD risk (Table 1) that are strongly correlated with lab-based LBD risk estimates (Figure 5B).

Adding pressure insoles to a wearable system does introduce additional implementation complexities and practical considerations. The current state of ergonomic assessment typically involves a safety professional with a clipboard and a tape measure, or a video camera (often still with oversight by a safety professional), or one or more IMU sensors worn by workers. Pressure insoles add some practical challenges relative to an IMU alone, for instance, related to needing a range of insoles that fit different shoe sizes and shapes and additional battery power and data storage demands. However, many commercial-off-the-shelf pressure insole products already exist, indicating that these challenges are manageable during product development and commercialization. Pressure insoles also add a few technological challenges. Most notably, pressure insole measurement accuracy and precision are common challenges [34]. Previous work has explored a variety of ways to improve pressure insole force estimates through advanced calibration techniques and processing methods, such as using transfer functions [35], neural networks [36], or multistage linear regressions [27]. Nonetheless, there continues to be room for innovation and advances in pressure insole calibration and signal processing to further improve force estimation accuracy and precision.

It is helpful to acknowledge that combining pressure insoles and a single IMU sensor is not technologically complex when compared with technologies commonly used in material handling environments, such as autonomous mobile robots, automated storage and retrieval systems, robotic arms, advanced forklifts, autonomous inventory robots, warehouse management systems, and wearable mobile computers. Likewise, from a biomechanics research and wearable sensor perspective, a two-sensor system is relatively sparse and simple compared with most alternative and previously used sensor sets [21,22,23]. Common challenges exist for all wearable sensor systems, including single IMU systems, such as wearability, ease of donning/doffing, fit (e.g., onto different body shapes), compatibility with work gear and environment, and consistent placement on the body. However, the recent growth and adoption of workforce wearable technologies indicates that these challenges can be managed [37]. Thus, we expect that there are various environments and use cases where Trunk+Force signals, such as obtained with an IMU and pressure insoles, would be practical to collect and could be beneficial in terms of automating ergonomic risk assessment.

### 4.3. Benefits and Drawbacks to Single IMU Systems

A single IMU system is simpler to don and doff, has lower battery power requirements, and does not need to accommodate different foot/shoe sizes as with pressure insoles. Several commercial wearable systems (e.g., StrongArm Fuse, Soter Analytics Clip&Go, Kinetic Reflex) take advantage of this relative simplicity using a single IMU mounted on the trunk or waist to provide useful insight and/or feedback on high-risk postures.

Matijevich et al. (2021) found that single trunk IMU systems can provide moderate accuracy for estimating time-series lumbar moments but tend to perform worse during higher musculoskeletal loading, which are often the instances of highest ergonomic interest. The current study results corroborate this limitation of trunk IMU systems by demonstrating the impact on LBD risk estimation. For instance, in Figure 5A, an LBD risk of 40% as estimated by the Trunk algorithm corresponded to ground truth (lab-based) LBD risks during a workday ranging anywhere from 17% to 78%. Thus, the Trunk system might identify two workdays as having similar risk when a worker experienced much different risks. Such large variance could potentially negate or limit the benefit of personalized, wearable monitoring of LBD risk using a single IMU.

This observation also highlights another issue: although machine learning and advanced algorithms receive much attention in the scientific literature and popular press, they are fundamentally limited by the signals they are given during training and use. Selecting an appropriate subset of sensors for a given application is paramount and provides the foundation on which successful algorithms are built and actionable insights are gleaned. This exemplifies why in our research progression (i.e., Matijevich et al., 2021, followed by this current study) we have initially focused our efforts on identifying the critical subset of sensor signals for LBD risk assessment while trying to balance accuracy and practicality of the wearable solution.

We dug deeper into the results and discovered that for lifting-intensive work, the main function and benefit of a trunk IMU for monitoring LBD risks may simply be estimating (counting) the number of bends or lifts performed each day. This is analogous to how a pedometer (step counter) counts steps per day as a rough proxy or indicator of physical activity, but without capturing or elucidating differences in speed or intensity (e.g., running vs. walking). In postprocessing, we found that the Trunk LBD risk estimates were strongly correlated with the number of lifts completed during each simulated workday (Figure 6B, r = 0.78) and weakly correlated with the average bend angle during lifting (Figure 6A, r = 0.33). To further explore the potential predictive value of a “lift-counter”, we reran the simulated workday analysis (detailed in *Methods*) but assumed that the damage from each lift was constant. This effectively assumed that every lift was of the same nominal weight (corresponding to a peak load moment of 75 Nm). This algorithm counted the number of lifts performed each workday, then computed the cumulative damage and LBD risk. With this “lift-counter” algorithm, we found an average LBD risk RMSE of 20% compared with lab-based estimates, which was only slightly higher than the 16% error seen when using Trunk signals.

### 4.4. Limitations

The limitations of this work should be considered and understood within the context that this study is one within a progression of experimental and computational studies. Collectively, these studies aim to develop science-based wearable sensor systems that are sufficiently accurate for remote ergonomic risk assessment and sufficiently practical to be translated into real-world use, ultimately to help reduce workplace injuries. Each individual study is limited in scope to address specific questions and to motivate next steps in research and technology development. Previously, Matijevich et al. (2021) identified the minimum sensor set (Trunk+Force) recommended to accurately estimate biomechanical load (time-series lumbar moment). This current study builds upon the prior work by evaluating the accuracy of this sensor set in terms of quantifying LBD risk over a full workday, which further informed and validated the potential of Trunk+Force wearable systems. Future work in this research progression is expected to implement and operationalize these algorithms in fully wearable systems (e.g., field-testable prototypes, commercial-grade products) to develop hardware-specific calibrations for pressure insoles and to evaluate algorithm performance during workplace studies or ergonomic assessments on workers.

The first limitation of this study is that we used idealized wearable signals to develop and evaluate the algorithms. Idealized wearable signals were beneficial for this early-stage research because they allowed us to explore the effects of different sensor signal combinations without being limited by the quality or accuracy of a single exemplary wearable sensor (e.g., a single product make/model) [26] or its default calibration. Actual wearable signals (from real wearable sensors) are expected to exhibit noise, errors, or drift that are not present in these idealized signals, but these can be managed or calibrated. In our previous work using wearable pressure insoles and an IMU on the foot to estimate tibial bone force during running [27], we initially obtained poor results from real wearable sensors and default (manufacturer) calibrations. Had we stopped there, we would have concluded that a system comprising a pressure insole and IMU could not accurately estimate tibial bone force. However, because we first analyzed idealized wearable sensor signals, we knew that the reason we were obtaining inaccurate results was not the combination of signals but rather the noise and variability in the real wearable sensor signals. We therefore created custom calibrations for the pressure insoles, which improved our estimation accuracy to a level close to that of idealized wearables [27] and ultimately led us to conclude that tibial bone forces can be monitored accurately using real, shoe-worn wearable sensors.

Second, there are other tools and models that can be used to evaluate LBD risk due to repetitive lifting, such as the NIOSH lifting equation. However, the NIOSH lifting equation [38] requires additional inputs that are more difficult to ascertain or collect with wearable sensors, such a coupling factor detailing the grip on the object. To capture this coupling factor would require a more complex system with additional sensors, which could be impractical to implement, and we wanted to maximize practicality for real-world use. LiFFT was therefore selected as a LBD risk assessment tool that we believed would be more directly compatible and implementable with wearable sensors and algorithms, and which has potential utility for the evaluation of other workforce wearable technologies, such as exoskeletons [24].

Third, the algorithms and simulations presented were for a finite subset of lifting tasks. However, these lifting tasks were relatively broad and selected because of their relevance to many material handling jobs (e.g., in warehouses, distribution, or fulfillment centers), where low back overexertion due to repetitive lifting is expected to be the dominant contributor to cumulative damage and fatigue of the lower back [21,39]. For other jobs where the weight of objects lifted, lifting techniques (symmetrical vs. asymmetrical) or frequency of lifts differs substantially from those evaluated in this study; then it may be beneficial to repeat workday simulation analysis or retrain LBD risk estimation algorithms using those different parameter ranges or lifting techniques.

Fourth, peak load moment was estimated based on the assumption that the object position and acceleration tracked with the motion of the participants’ trunk (Section 2.3). To assess whether this was reasonable and appropriate, we performed two sub analyses. One is presented in Appendix A, where we reran our analysis using lumbar moment instead of load moment. The lumbar moment calculation does not include any assumptions about box motion, yet this analysis resulted in the same main findings and conclusions as when using load moments. Next, we performed a supplemental analysis on a subset of lifting data using the position and acceleration of the hands as a surrogate for the box’s motion. Thus, we did not assume the box distance and acceleration tracked with the trunk. We found that using hand motion resulted in a minimal change (<5%) in peak load moment, as compared with using the approach summarized in Section 2.3. Collectively, these additional analyses suggest that the assumptions used to compute load moment in this study were reasonable, since even when they were removed, they had minimal effect on load moment estimates and did not alter the main conclusions.

## 5. Conclusions

In conclusion, combining trunk motion with under-the-foot force data markedly improved the accuracy of LBD risk estimates compared with using only trunk motion data. These results indicate that a wearable system composed of a single IMU on the trunk, pressure-sensing insoles in shoes, and trained algorithms could be valuable for remote and/or automated ergonomic assessment of low back injury risks during material handling. The results also raise fundamental questions and concerns about the adequacy of using trunk motion data alone for LBD risk assessment during material handling.

## Figures and Tables

**Figure 1 sensors-23-02064-f001:**
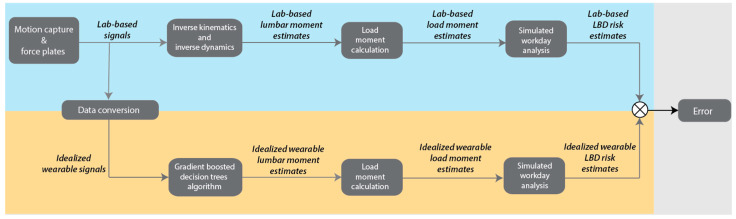
Lab-based and idealized wearable LBD risk analysis and evaluation overview. This workflow shows how data were processed to estimate LBD risk from the lab-based and idealized wearable (Trunk and Trunk+Force) signals. Blue (upper track) represents the lab-based signal analysis. Orange (lower track) represents the idealized wearable signal analysis, which was completed independently for Trunk and Trunk+Force signals. This workflow was first performed using only idealized Trunk signals to develop an algorithm and compute error. Then the workflow was repeated for idealized Trunk+Force data.

**Figure 2 sensors-23-02064-f002:**
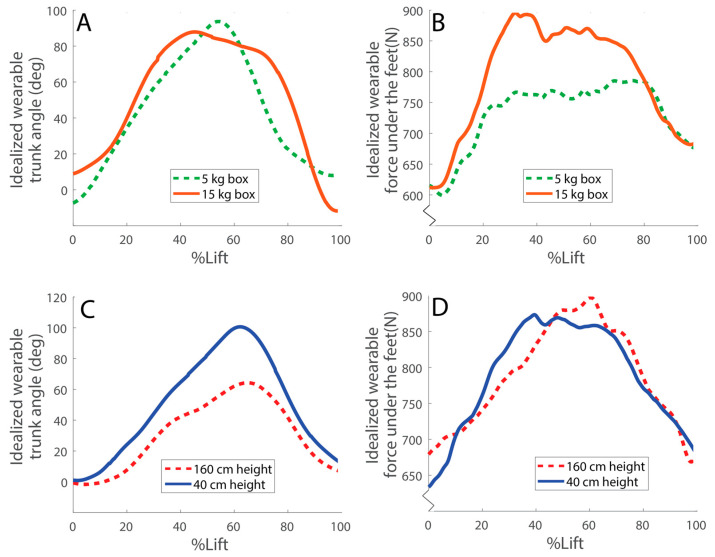
Examples of idealized wearable Trunk and Force data. Shown are examples of lifting tasks performed by a representative participant. The top row of plots shows two trials: one lifting 5 kg and one lifting 15 kg, both from the same height off the ground. Peak trunk angle is similar (**A**), but the force under the feet is higher for the heavier object in these trials (**B**). The bottom row of plots shows two trials: both involve lifting an object of the same weight, but one is lifting from 40 cm and one from 160 cm above the ground. Peak trunk angle is larger when lifting from a lower height (**C**), whereas peak force under the feet is similar in these trials (**D**).

**Figure 3 sensors-23-02064-f003:**
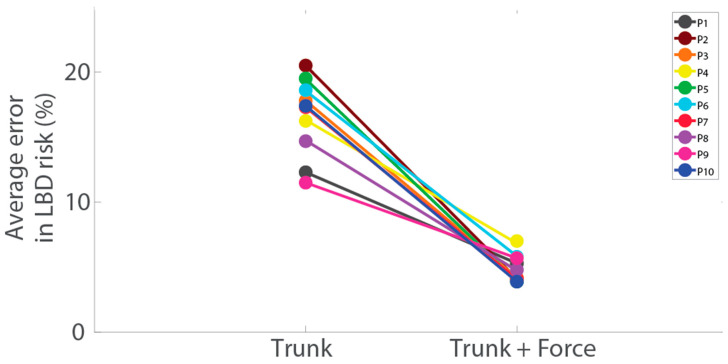
RMSE of the LBD risk calculated from Trunk and Trunk+Force signals compared with the LBD risk obtained from lab-based estimates. Each color represents a different participant (P). Lines are shown to easily visualize differences for each participant.

**Figure 4 sensors-23-02064-f004:**
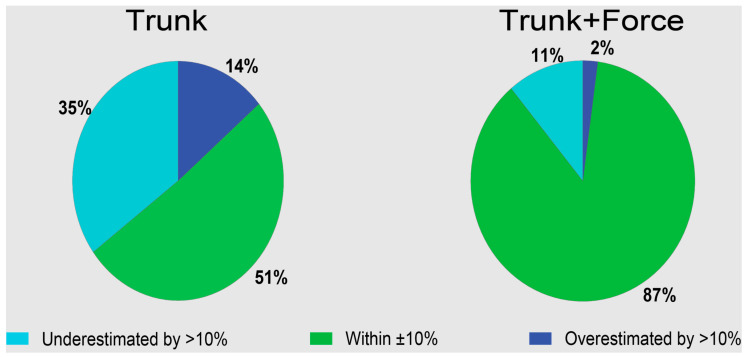
The percentage of simulated workdays that had LBD risk estimates within ±10 of the lab-based estimates (green) vs. overestimated (dark blue) or underestimated (light blue) by >10%.

**Figure 5 sensors-23-02064-f005:**
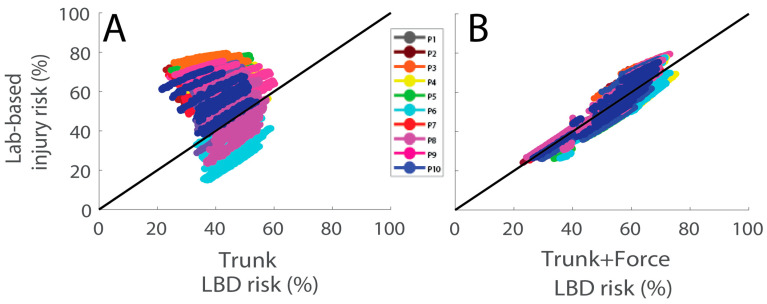
LBD risk estimates from (**A**) Trunk (r = 0.48) and (**B**) Trunk+Force (r = 0.97) each vs. lab-based estimates of LBD risk. Each color represents a different participant (P), and each data point a different simulated workday. The black line is a unity line representing perfect estimates of LBD risk and is provided for visual reference. Participant-specific correlation coefficients are provided in Table 3.

**Figure 6 sensors-23-02064-f006:**
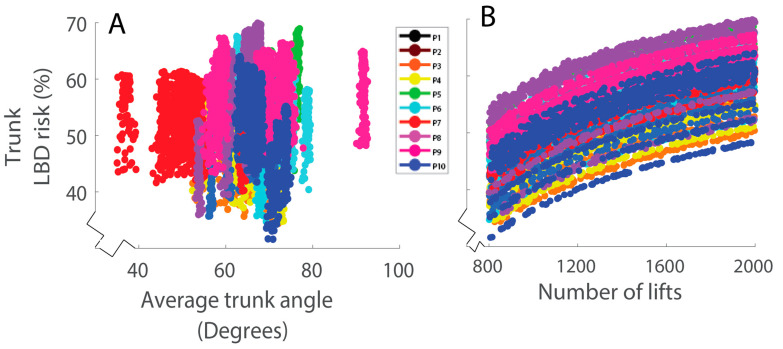
LBD risk estimates from Trunk signals are plotted as a function of (**A**) average trunk bend angle and (**B**) number of lifts during each simulated workday. The number of lifts performed in a simulated workday was strongly correlated (r = 0.78) with the Trunk LBD risk estimate of that day, whereas the average bend angle was weakly correlated (r = 0.33). Each color represents a different participant (P), and each data point a different workday.

**Table 1 sensors-23-02064-t001:** Participant-specific results showing errors (RMSE) in LBD risk estimates during simulated workday analysis. RMSE is shown for Trunk and Trunk+Force estimates compared with lab-based estimates.

RMSE in LBD Risk Estimates		
Participant	Trunk	Trunk+Force
1	12.3%	5.3%
2	20.7%	4.1%
3	15.8%	4.2%
4	14.3%	7.4%
5	19.9%	3.9%
6	18.5%	5.8%
7	16.3%	4.1%
8	13.7%	4.8%
9	11.5%	5.7%
10	16.4%	3.9%
**Avg**	**15.8 ± 2.3%**	**4.7 ± 1.1%**

**Table 2 sensors-23-02064-t002:** Participant-specific accuracy results showing how often LBD risk estimates from Trunk and Trunk+Force estimates were within 10% of lab-based LBD risk estimates.

Within 10% of Lab-Based LBD Risk		
Participant	Trunk	Trunk+Force
1	59.9%	90.0%
2	37.8%	86.9%
3	59.5%	90.7%
4	39.5%	86.0%
5	44.2%	86.2%
6	39.8%	84.2%
7	64.8%	78.5%
8	40.6%	90.3%
9	64.9%	89.7%
10	58.5%	89.4%
**Avg**	**50.9 ± 11.5%**	**87.2 ± 3.8%**

**Table 3 sensors-23-02064-t003:** Participant-specific results showing the correlation between LBD risk estimates from Trunk and Trunk+Force estimates and the lab-based LBD risk estimates. Average correlation coefficients were computed by transforming Pearson correlation coefficients (r) for each participant to Fisher z scores. The average Fisher z score was computed and then transformed back into a Pearson correlation coefficient for reporting.

r Compared with Lab-Based LBD Risk		
Participant	Trunk	Trunk+Force
1	0.41	0.93
2	0.20	0.96
3	0.56	0.98
4	0.49	0.97
5	0.50	0.94
6	0.42	0.95
7	0.53	0.96
8	0.56	0.94
9	0.55	0.98
10	0.45	0.93
**Avg**	**0.48**	**0.97**

## Data Availability

Data is available by request.

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
