# Peer review of "How Accurately Can Wearable Sensors Assess Low Back Disorder Risks during Material Handling? Exploring the Fundamental Capabilities and Limitations of Different Sensor Signals"

_sensors, 2023, doi:10.3390/s23042064_

Round 1

Reviewer 1 Report

As in the attached pdf file

Author Response

Response to reviewer 1 can be found in the attached document 

Reviewer 2 Report

Thank you very much for the opportunity of this very interesting article.

I find the article interesting and novel but I think the focus and development of the algorithms are precarious. First of all, they develop calculation methods full of approximations, understandable since this is a study that focuses on reducing the number of technologies to analyze the risk of musculoskeletal injuries, but what difference is there between the Trunk and Trunk+insoles data if in the calculation methods shown in subsection 2.3 all variables are calculated using the IMU, anthropometric tables or approximations? If there are two types of data groupings (log and log+templates) for the input, why are the methods of the algorithms, LiFFT or simulated days exactly the same?

Both methods and results must be modified and clarified for the paper to be published. I also do not understand why they have not taken data from 10 new participants with a single IMU and pressure templates replicating the experimental conditions of the previous study, and later compared it with the previous data (laboratory conditions).

In addition, here are some recommendations.

Introduction

-I would add a sentence explaining with which technology or how the LiFFT instrument was validated.

-Line 54 and 64: I would rewrite this paragraph as you are advocating for a simplified intervention but that involves the use of IMUs and insoles, therefore advocating that individualized interventions goes against your own. 

-From line 57 to 77 there is no bibliographic reference, please justify your assertions.

Methods

-Line 182-183, if the study was approved by a Bioethics Committee as you mention, please provide the Bioethics Committee code or number. Also add the corresponding section at the end of the manuscript.

-Line 208-217, I think this paragraph is one of the most important in the article. However, they make a series of approximations without any reference to justify them with previous studies. Similarly, comparing data obtained with a force platform and by a pressure template without prior validation is risky.

-Line 242, the reference is not correctly formatted.

-Load moment calculation methods cannot be considered appropriate. If you intend to compare the results obtained by the algorithm with the LiFFT instrument, how can you assume that the distances (L) for calculating Mlineal are the same, when the LiFFT scale does take this distance between object and user as a determinant in its parameters.  Similarly, assuming that the distance from the object to the lumbar spine and from the CoM HAT to the lumbar spine is the same is simplistic at best. Likewise, assuming that the acceleration of the upper body and the lifted object is the same may be highly questionable. 

Results

-I would replace Figure 4 with a point cloud around the perfect relationship (black line).

Discussion

-Without comment, both the discussion and conclusions depend on the previous sections.

Author Response

Response to reviewer 2 can be found in the attached document 

Reviewer 3 Report

The authors used two data-driven simulations to test whether it is possible to estimate the risk of LBD injury. Their results are interesting. However, I have a several questions:

1.       I'm not sure what studies you have done. Did you do an analysis across 5 pieces? How about inserting a figure that gives an overall picture of the study?

2.       How was the risk of LBD calculated using outcomes?

3.       It is my understanding that the analysis of moments in floor reaction form is used primarily in the lower extremities. Is there any validity or reliability in the values of trunk moments?

Author Response

Response to reviewer 3 can be found in the attached document

Round 2

Reviewer 1 Report

Dear Editor

The authors have addressed some of the reviewer comments in the first round. However, some of my previous comments are still not addressed as follows:

1.       A piece of the collected data of the Trunk and Trunk+Force from participants must be provided as a figure or table.

2.       The legends of Figures 2 and 5 are still not provided. The reader prefers to see a legend for figures, especially those in current work, including more colors.

3.       Correlation coefficients must be provided in Figures 4A and 4B.

4.       Still, some references are outdated, such as refs 5 (year of publication 1985), 10 (year of publication 2008), 12 (year of publication 1997), 13 (year of publication 1993), and 24 (year of publication 1996). New ones must replace these.

Best regards,

Reviewer

Author Response

Response to reviews are attached

Reviewer 2 Report

The authors have improved and clarified several issues in section 2.2. regarding the calculation algorithms.

I really understand the need to do some preliminary work to open the very interesting line of research that you propose, but in my humble opinion you use and manage cutting-edge technology and complex algorithms, that is why I am surprised by the use of the LIFFT scale (full of approximations and reductions).

Introduction

All corrections are optimal and follow the narrative of the article.

Material and Methods

The justification of the ethical issues and the use of pressure insoles has been correctly solved.

Although the simplifications of the anthropometric tables in the calculation methods persist, the authors have shown a very appropriate intention linked to the scientific method. They have justified each of the calculations in different sections, for which I can only congratulate them for their work and thank them.

Results

As this is a minor correction and taking into account the extensive and beneficial corrections of the authors included in the article, the figure remains the same showing relevant information.

Discussion and Conclusions

As with the entire article, they have improved substantially.

Congratulations to the whole team, the work developed and the corrections provided say a lot about your research team. Keep up the good work

Author Response

Response attached. Thank you for your kind words

Reviewer 3 Report

Thank you for your correction.
